# Endosomal Escape and Nuclear Localization: Critical Barriers for Therapeutic Nucleic Acids

**DOI:** 10.3390/molecules29245997

**Published:** 2024-12-19

**Authors:** Randall Allen, Toshifumi Yokota

**Affiliations:** 1Department of Medical Genetics, Faculty of Medicine and Dentistry, University of Alberta, Edmonton, AB T6G 2H7, Canada; 2The Friends of Garrett Cumming Research & Muscular Dystrophy Canada HM Toupin Neurological Sciences Research, Edmonton, AB T6G 2H7, Canada

**Keywords:** therapeutic nucleic acids, antisense oligonucleotides, small interfering RNA, endosomal escape, nuclear localization, drug delivery

## Abstract

Therapeutic nucleic acids (TNAs) including antisense oligonucleotides (ASOs) and small interfering RNA (siRNA) have emerged as promising treatment strategies for a wide variety of diseases, offering the potential to modulate gene expression with a high degree of specificity. These small, synthetic nucleic acid-like molecules provide unique advantages over traditional pharmacological agents, including the ability to target previously “undruggable” genes. Despite this promise, several biological barriers severely limit their clinical efficacy. Upon administration, TNAs primarily enter cells through endocytosis, becoming trapped inside membrane-bound vesicles known as endosomes. Studies estimate that only 1–2% of TNAs successfully escape endosomal compartments to reach the cytosol, and in some cases the nucleus, where they bind target mRNA and exert their therapeutic effect. Endosomal entrapment and inefficient nuclear localization are therefore critical bottlenecks in the therapeutic application of TNAs. This review explores the current understanding of TNA endosomal escape and nuclear transport along with strategies aimed at overcoming these challenges, including the use of endosomal escape agents, peptide-TNA conjugates, non-viral delivery vehicles, and nuclear localization signals. By improving both endosomal escape and nuclear localization, significant advances in TNA-based therapeutics can be realized, ultimately expanding their clinical utility.

## 1. Introduction

After several decades of intensive research, therapeutic nucleic acids (TNAs) such as antisense oligonucleotides (ASOs) and small interfering RNA (siRNA) have emerged as next-generation therapeutic strategies for a wide range of diseases. ASOs are single-stranded, synthetic molecules, typically 15–30 nucleotides in length, designed to modulate gene expression [1]. ASOs function by hybridizing to a target mRNA strand through Watson–Crick base pairing, leading to one of two primary mechanisms of action: RNase H-mediated degradation or steric hindrance. In the first mechanism, ASO-mRNA duplexes are recognized by the enzyme RNase H1, which directs the mRNA for degradation [2]. Alternatively, ASOs can provide a steric blockage which prevents the binding of proteins to the target mRNA. Such blockages can have multiple effects including translation repression or the modulation of pre-mRNA splicing [3,4]. Another commonly explored nucleic acid therapeutic can be found in siRNA which are short, double-stranded RNA molecules functioning through the RNA interference (RNAi) mechanism. Following siRNA cleavage, interactions with the RNA-induced silencing complex (RISC) promotes the degradation of target mRNA [5]. Similarly to RNase H-mediated degradation, siRNA can be utilized in a therapeutic context to downregulate the expression of a disease-causing protein. Regardless of the specific mechanism of action, all TNAs present a major advantage in their ability to target traditionally “undruggable” proteins. While only around 2% of the human proteome is accessible to small molecule drugs, TNAs can be designed to target virtually any gene, requiring only the mRNA target sequence [6]. This versatility has fueled significant investment in TNA research and commercialization.

Like any therapeutic innovation, TNAs face certain limitations. Oral administration is inefficient, necessitating alternative delivery methods such as subcutaneous injection or intravenous infusion [7]. After injection, up to 40% of TNAs tend to accumulate in the liver, while another significant portion are rapidly filtered by the kidneys and excreted in the urine [8,9]. Due to their nucleic acid-like structure, TNAs are also susceptible to degradation by nuclease enzymes, both in the circulation and following cellular entry [10]. Direct delivery to target tissues, such as intravitreal or intrathecal injections, can mitigate some of these challenges, though these routes often carry additional risks and potential adverse effects. To address these challenges, extensive research has focused on structural modifications to enhance TNAs’ resistance to nuclease degradation, improve their solubility, and increase their overall therapeutic efficacy [11].

Although great progress has been made in the field of nucleic acid therapeutics, due to their mechanism of cellular uptake, several key limitations persist. TNAs are generally large, negatively charged molecules, which prevents them from passively diffusing across the cell membrane. Although the precise mechanisms of uptake are not fully understood, endocytosis is known to play a major role. Following binding to cell surface proteins, TNAs are typically internalized through clathrin- or caveolin-mediated endocytosis [12]. These pathways are commonly referenced in TNA research but are by no means exclusive. Studies using clathrin and caveolin inhibitors have demonstrated minimal impact on TNA internalization, suggesting that other mechanisms are also involved [13]. Additional pathways, such as macropinocytosis and the CLIC/GEEC pathway, have been reported to contribute to TNA uptake. Once internalized, TNAs are encapsulated in membrane-bound structures called endosomes. These early endosomes undergo a maturation process into multivesicular bodies (MVBs) and late endosomes from which several intracellular trafficking routes can occur. One common pathway involves the fusion of late endosomes with lysosomes, which serve as the cell’s primary degradation centers [14]. Other pathways include transport to autophagosomes or the Golgi apparatus, both of which have been implicated in TNA trafficking [15,16]. Alternatively, TNAs can be directed to recycling endosomes, which return to the cell surface, or retained in stable structures known as “depot” endosomes, leading to long-term cellular storage [17,18] (Figure 1). The major challenge across all trafficking routes is the retention of TNAs within membrane-bound vesicles. Whether degraded in lysosomes, trapped in endosomes, or recycled back out of the cell, TNAs are largely prevented from reaching their target mRNA and exerting a therapeutic effect [19]. This is known as the “endosomal escape problem”, which is a major barrier in effective delivery. Among various parameters affecting TNA efficacy, endosomal escape has been shown to correlate most strongly with transfection efficiency, underscoring its critical role in successful delivery [20]. In fact, studies suggest that only 1–2% of exogenously administered TNAs manage to escape endosomal entrapment and reach the cytosol [21].

Endosomal escape, while it is a significant bottleneck, is not the final hurdle in delivery for all TNAs. After escaping endosomal entrapment, TNAs freely diffuse through the cytoplasm with only a portion successfully reaching the nucleus for internalization [22]. Accumulation within the nucleus is of particular importance for splice switching oligonucleotides (SSOs) which modulate pre-mRNA splicing. Additionally, both RNase H-mediated degradation and RNAi mechanisms act on nuclear along with cytoplasmic targets [23,24]. Therefore, efficient nuclear localization following endosomal escape represents another critical barrier to TNA efficacy.

Although these delivery barriers have long been recognized, overcoming them remains a significant challenge. Given the clinical success of approved TNAs, even small improvements in endosomal escape or nuclear localization could lead to significant gains in therapeutic efficacy. This review aims to provide an overview of the current understanding of these processes, alongside innovative approaches that could pave the way for improved delivery systems. Continued exploration in these areas holds the promise of overcoming existing limitations, ultimately advancing the development of TNA-based therapies and expanding their applicability in treating a broader range of conditions.

## 2. Endosomal Escape and Nuclear Localization of TNAs

It is evident that some natural mechanisms of endosomal escape must occur, as freely delivered TNAs can reach their intracellular targets, exerting a therapeutic effect. However, the precise processes by which TNAs naturally escape from endosomes and localize to the nucleus remain poorly understood. One theory suggests that biological membranes, including endosomal membranes, undergo spontaneous fluctuations [25]. These fluctuations may create temporary openings in the membrane, which allow for escape. The random and transient nature of these events, with the rapid resealing of the membrane, could explain why only a small fraction of TNAs successfully escape endosomal entrapment.

Another proposed mechanism for endosomal escape involves membrane fusion events. During endosomal maturation, MVBs form, consisting of numerous intraluminal vesicles (ILVs) [26]. The process of ILV formation requires the endosomal membrane to pinch off, creating an opportunity for the intraluminal contents, including TNAs, to escape. Additionally, these ILVs can undergo a process known as retrofusion, in which they re-fuse with the limiting membrane of the MVB [27]. The hypothesis that such fusion events contribute to endosomal escape is supported by data indicating that the majority of endocytosed therapeutics are released from intermediate endosomal compartments like MVBs [21,28]. There is a general consensus that early escape in the trafficking process is crucial for effective cytosolic release, while TNAs that are retained are likely to be degraded in lysosomes. However, visualizing such rapid escape events at the nanometer scale is technically challenging, resulting in limited empirical evidence to support these hypotheses. Consequently, the exact location and mechanisms of endosomal escape remain unresolved questions.

Recent evidence highlights the importance of endosome–Golgi transport pathways in facilitating endosomal escape. For instance, studies have shown that following the administration of free ASOs, COPII-coated vesicles, which are typically involved in the anterograde transport of proteins from the endoplasmic reticulum (ER) to the Golgi apparatus, exhibit a significant association with ASO-containing LEs [29]. Moreover, the knockdown of COPII tethering proteins has been shown to impair endosomal escape, supporting the hypothesis that the binding of these vesicles to endosomes could enhance the release of ASOs. Further investigations have identified Golgi-derived vesicles containing the mannose-6-phosphate receptor (M6PR) as crucial players in this process [30]. M6PR vesicles are primarily responsible for transporting hydrolase enzymes to the endo-lysosomal system; it is posited that fusion events between these vesicles and the endosomal membrane may facilitate escape.

The few TNAs that successfully escape endosomal entrapment disperse throughout the cell before encountering another barrier: the nuclear membrane. Some oligonucleotides have been observed to freely diffuse into the nucleus, likely leveraging their relatively small size to pass through the nuclear pore complex [31]. However, nuclear import is a highly regulated process influenced by factors such as TNA size, charge, and secondary structure. For instance, decreasing temperature has been shown to inhibit the nuclear transport of phosphorothioate (PS)-modified ASOs more significantly than their unmodified counterparts [32]. This finding suggests that the modified ASOs may rely on active transport mechanisms to traverse the nuclear pore complex. Other exogenous RNA molecules have been reported to enter the nucleus via interactions with nuclear shuttle proteins, such as importins, and a similar active transport mechanism may be utilized by TNAs [33].

While the potential for TNAs to effectively reach their intracellular targets is evident, the precise mechanisms underlying their natural escape from endosomes and their subsequent localization to the nucleus remain poorly understood. In reality, a combination of different mechanisms may be at play, depending on the specific cell type and cargo involved. Given the complexity and variability of these processes, continued research is essential to elucidate the multifaceted mechanisms that govern TNA trafficking and localization.

## 3. Traditional Endosomal Escape Strategies

Given the significant challenge that endosomal escape presents for the delivery of TNAs, extensive research has focused on strategies to enhance this critical process. Many delivery approaches exploit materials that respond to endogenous or exogenous stimuli to trigger endosomal escape events. Endosomal compartments experience a stepwise reduction in pH, with values decreasing from approximately 6.5 in early endosomes to around 4.5 in lysosomes [34]. Many strategies leverage this characteristic by utilizing pH-responsive materials that remain inert in the extracellular environment (pH 7.0) but activate their escape mechanisms specifically within endosomal compartments. In addition to pH sensitivity, endosomal escape can also be facilitated by exogenous stimuli such as light, magnetic fields, or ultrasound, which can induce membrane destabilization [35]. This section will review the mechanisms and advancements of these traditional endosomal escape strategies, including endosomal-disrupting small molecules, proteins/peptides, and non-viral delivery vehicles (Figure 2).

### 3.1. Endosomal Disrupting Small Molecules

Small molecules offer significant advantages over other biologic delivery enhancers in terms of solubility, stability, and the potential for oral administration. As a result, there has been considerable interest in identifying small molecule endosomal escape enhancers. One notable example is the antimalarial drug chloroquine. At cytosolic pH levels, chloroquine readily enters cells, but becomes protonated in the acidic environment of endosomes due to its weakly basic properties [36]. This protonation increases osmotic pressure and destabilizes the endosomal membrane, a phenomenon known as the proton sponge effect, also referred to as the pH buffering effect, osmotic swelling, or osmotic lysis. The underlying mechanism of the proton sponge effect is linked to the activity of the V-ATPase, a membrane proton pump in endo/lysosomes which maintains the intraluminal pH at a specific level [37]. When pH buffering molecules enter, the V-ATPase compensates by pumping more protons into the vesicle. This influx of protons is accompanied by the entry of chloride ions and water, significantly increasing osmotic pressure and creating openings in the membrane for escape [38]. These characteristics have made chloroquine a prototypical small molecule endosomal escape enhancer. Other cationic amphiphilic drugs (CADs), such as siramesine and amitriptyline, have also been shown to enhance the delivery of freely delivered siRNA, presumably through a similar mechanism [39]. However, a significant drawback of these CADs is that the resulting endosomal membrane rupture can induce unacceptable levels of cellular stress.

Despite toxicity limitations, further exploration of potential small molecule endosomal escape enhancers has not been deterred. Researchers at the University of North Carolina conducted a screening of small molecules from their compound library of UNC molecules for the ability to enhance TNA delivery [40]. Notably, one compound, UNC7938, was identified as significantly increasing both ASO and siRNA activity by promoting their release from endosomal compartments. Follow-up studies confirmed that UNC7938 also enhances in vivo treatment efficacy following the intravenous injection of a freely delivered exon-skipping ASO in the Mdx mouse model [41]. The mechanism of action for UNC7938 involves destabilizing the endosomal membrane, differentiating it from CADs like chloroquine, which primarily rely on increasing osmotic pressure [42]. While UNC7938 has demonstrated promise, safety assessments have yielded mixed results; some studies reported a 90% cell death rate in vitro, whereas no evidence of liver or kidney toxicity was observed in vivo [41,43]. In contrast, the less potent small molecule UNC4267 has also been shown to enhance endosomal escape with fewer toxic effects [44]. However, this reduced toxicity correlates with decreased membrane disruption and, consequently, less effective endosomal escape. It appears that enhancing endosomal escape with such drugs necessitates a certain degree of toxicity, largely relegating their application to in vitro transfection enhancers.

### 3.2. Proteins and Peptides

Early research into protein-mediated endosomal escape primarily focused on endolytic peptides that induce membrane destabilization in response to the low pH of endosomes [45]. However, these treatments encounter similar toxicity limitations as those seen with endolytic small molecules. To mitigate toxicity concerns, alternative peptide-based strategies have been developed. This section will review two promising peptide platforms: biomimetics and cell-penetrating peptides (CPPs), which can effectively target endosomal membranes and promote the release of therapeutic payloads into the cytosol.

#### 3.2.1. Biomimetics

To enhance endosomal escape, some of the most efficient models can be found in bacterial toxins and viruses, which enter cells via endocytosis but require escape from endosomes to exert their effects. One notable mechanism utilized by bacterial toxins is the action of pore-forming proteins. In the acidic environment of endosomes, bacterial toxins can create pores that facilitate escape into the cytosol [46]. Researchers have explored models based on these proteins, demonstrating that mutant forms of the listeriolysin O toxin (LLO) significantly enhance endosomal escape when conjugated to the surface of gold nanoparticles [47]. While these proteins are effective at perforating endosomes, they can also form pores in the plasma membrane, potentially causing significant cellular damage. To mitigate such negative effects, innovative strategies like co-administration with toxin-neutralizing antibodies have been investigated [48]. This approach employed a perfringolysin O (PFO)-antibody conjugate to prevent toxin activity in the cytoplasm, allowing PFO to be released only after endocytosis, increasing the therapeutic window five-fold.

Other toxin-mimetics, such as modified diphtheria, shiga, and botulinum toxins, have been engineered for intracellular delivery with various strategies to reduce toxicity [49,50,51]. Toxin proteins are often composed of functional domains, each with specific roles such as toxicity or membrane fusion. By selectively targeting certain domains, researchers can achieve their desired effects. For example, in an attenuated diphtheria toxin conjugated to siRNA molecules, only the active domain responsible for toxicity was modified [49]. This approach preserved effective uptake and endosomal escape by retaining the translocation and receptor binding domains, while significantly reducing toxicity. Alternatively, toxic domains can be completely removed. Park et al. focused solely on the endosomal escape domain of botulinum neurotoxin and conjugated it to their protein delivery system [51]. This strategy resulted in a 100-fold increase in cytosolic delivery, attributed to enhanced endosomal escape.

Enveloped viruses, such as influenza A, utilize a membrane fusion mechanism for endosomal escape. This process is mediated by the hemagglutinin (HA) glycoprotein on the viral surface [52]. The acidic pH of endosomes triggers a conformational change in HA, initiating fusion with the endosomal membrane [53]. In contrast to the approximately 1% escape rate observed for TNAs, HA has been estimated to achieve an escape efficiency in the range of 30–70% [18]. Given the impressive escape capabilities of such proteins, researchers have begun exploring biomimetic versions of this protein. Dr. Steven Dowdy’s lab is currently developing a synthetic HA-like delivery system aimed at creating a universal endosomal escape domain (uEED) [54]. This system utilizes a hydrophilic outer domain to mask the internal membrane fusion domain. Upon endosomal entry, the low pH environment prompts the shedding of the masking domain, allowing for endosomal escape. A significant advantage of this approach is that synthetic proteins do not elicit the same immunogenic response as natural viral proteins. Moreover, promoting fusion with the endosomal membrane reduces the risk of membrane rupture, thus limiting toxicity.

Overall, proteins derived from bacterial toxins and viruses demonstrate remarkable efficiency in endosomal escape. However, the toxicity associated with pore formation and potential adverse immune responses remain significant challenges for clinical drug delivery applications. Ongoing research continues to develop strategies aimed at minimizing these negative side effects.

#### 3.2.2. Cell-Penetrating Peptides

Cell-penetrating peptides (CPPs) are short peptide sequences that, as their name suggests, can be readily internalized by cells. Most CPPs are rich in positively charged amino acids, such as arginine and lysine, which enhance their interaction with the negatively charged plasma membrane [55]. While the positive charge of CPPs facilitates cellular uptake, it can complicate TNA delivery due to their negative charge which causes the formation of TNA-CPP aggregates. Therefore, this strategy is more commonly used for neutrally charged molecules, such as phosphorodiamidate morpholino oligonucleotides (PMOs), a type of ASO. Indeed, CPP-PMO conjugates have been shown to significantly enhance ASO therapeutic efficacy both in vitro and in vivo [56].

Despite their ability to improve the delivery of therapeutics, the cellular uptake process of CPPs still predominantly relies on endocytosis, resulting in endosomal entrapment. Therefore, additional strategies are necessary to promote endosomal escape for these peptides. A key feature in designing CPPs for endosomal escape involves leveraging the natural pH gradient found within the endosomal pathway. Such peptides can exist as random coils at a physiological pH but adopt an alpha-helical structure in the acidic environment of endosomes, facilitating membrane penetration [57]. Other factors, including the histidine–arginine ratio, hydrophobicity, and helicity, are also crucial determinants of a peptide’s escape ability [58,59,60]. One of the most well studied CPPs is TAT, a positively charged protein derived from the HIV virus [61]. Notably, the protein transduction domain (PTD) of TAT is recognized for its strong ability to facilitate cell entry. A dimerized form of TAT (dfTAT), linked by a disulfide bond, is believed to significantly enhance endosomal escape by disrupting the endosomal membrane [62].

Non-canonical cyclic CPPs have also emerged as promising endosomal escape agents. For instance, cFΦR4, a cyclic variant of an arginine-rich CPP, has been shown to significantly enhance cytosolic delivery compared to its linear counterparts [63]. Importantly, these cyclic peptides were found to escape from early endosomal compartments, limiting the potential degradation of their cargo by enzymes. Other studies on cyclic CPPs confirmed their capacity for high cytosolic delivery, although there was no evidence of endosomal rupture, suggesting a mechanism alternative to membrane disruption [64]. Using model membrane systems and confocal microscopy, researchers visualized the formation of small, distinct budding vesicles. This led to a proposed mechanism whereby protein–membrane interactions induce the creation of miniature vesicles following endocytosis. These small vesicles, once formed, are unstable; upon release, a loss of their pH gradient causes them to collapse and release their contents into the cytosol. Subsequent studies corroborated the vesicle budding and collapse hypothesis in other CPPs and the diphtheria toxin [65,66]. Such peptides improved cytosolic delivery by up to 120% compared to the mere 2% observed with traditional CPPs like TAT. While cyclic CPPs were originally investigated for protein delivery, they have recently been validated in the context of oligonucleotide conjugation, enhancing the efficacy of SSOs administered to mouse models of two neuromuscular disorders: duchenne muscular dystrophy and facioscapulohumeral muscular dystrophy [67].

While CPPs certainly increase TNA efficacy, whether this enhancement arises primarily from improved endosomal escape or through other mechanisms remains a complex question. In addition to endocytosis, CPPs may also enter cells via direct translocation through the plasma membrane [68]. Reports indicate that CPPs, traditionally thought to enhance endosomal escape, show negligible cytosolic delivery when this direct translocation pathway is inhibited by membrane depolarization [69]. This finding suggests that endocytosis followed by endosomal escape may not be the primary productive mechanism for TNA delivery via CPPs. Further studies employing a Split Luciferase Endosomal Escape Quantification (SLEEQ) assay have indicated that endosomal escape is not significantly enhanced using CPPs [70]. One hypothesis for these findings is that CPPs may merely facilitate endocytosis without enhancing endosomal escape, meaning the delivery benefits of CPPs could stem simply from increasing cellular uptake. Alternatively, a portion of peptides capable of directly translocating the plasma membrane may be responsible for the increased therapeutic efficacy. Ultimately, evidence is quite mixed on the specific uptake pathways of CPPs and how they improve delivery. The precise mechanisms may vary based on the specific CPP in question and the cargo it is delivering.

### 3.3. Non-Viral Delivery Vehicles

An increasingly popular strategy to enhance endosomal escape involves the use of non-viral delivery vehicles. Unlike viral delivery systems, such as AAV9, which are frequently employed for transgenic applications, non-viral delivery vehicles offer a promising alternative. Viruses naturally possess endosomal escape mechanisms that facilitate the delivery of genetic material to the cytosol, making them a logical choice for therapeutic delivery. However, concerns regarding toxicity and immunogenicity associated with viral vectors have limited their practical applications [71]. Considering these limitations, research has shifted significantly toward non-viral methods. Non-viral delivery vehicles can improve the stability and cellular uptake of TNAs while also being engineered for enhanced endosomal escape [72]. This section will explore several innovative non-viral delivery formulations specifically engineered to enhance endosomal escape, focusing on lipid-based systems and exogenous stimuli-responsive vehicles.

#### 3.3.1. Lipid Delivery Vehicles

Lipid nanoparticles (LNPs) are widely used for the delivery of nucleic acids such as in mRNA vaccines, but they still face the challenge of poor cytosolic delivery, like freely delivered TNAs [21]. To address this limitation, researchers have focused on modifying LNP systems to improve endosomal escape. A key approach involves the use of cationic lipids, which enhances cellular uptake and facilitates interactions with the negatively charged endosomal membrane [73]. This interaction triggers a membrane fusion mechanism in which the lipids transition from a lamellar to an inverted hexagonal phase, promoting fusion with the endosomal membrane and enabling cargo release into the cytosol [74]. However, cationic lipids also pose toxicity risks, potentially initiating apoptotic and inflammatory pathways [75].

As a result, much of the research has shifted toward using ionizable LNPs. These contain lipid head groups that remain neutral at a physiological pH but become protonated in the acidic environment of the endosome. This property allows them to stay inert during delivery and cellular uptake, but actively associate with the endosomal membrane to promote escape once inside the endosome. Nanoparticles composed of ionizable lipids such as DLin-MC3-DMA and ALC-0315 have demonstrated increased transfection efficacy, likely due to enhanced endosomal escape [76]. This type of system can reduce the cytotoxicity seen with traditional cationic lipids while retaining high delivery efficiency.

Additionally, neutral lipids can help mitigate toxicity concerns when used in conjunction with cationic lipids. For instance, DNCA/CLD, a widely studied platform, combines the neutral cytidinyl-lipid DNCA with the cationic lipid CLD to deliver TNAs effectively [77]. Beyond electrostatic interactions, hydrogen bonding and pi-stacking between lipids and nucleic acids facilitate TNA complexation into nanocomplexes. In one study, fluorescently labeled mRNA delivered via DNCA/CLD showed cytosolic localization near late endosomes/lysosomes, suggesting potential for endosomal escape [78]. Moreover, DNCA/CLD demonstrated no significant cytotoxicity compared to non-treated controls across three different cell lines, unlike the cationic lipofectamine, which caused substantial cell death. These findings highlight potential for the integration of neutral lipids in reducing toxicity while maintaining efficient TNA delivery.

Anionic LNPs offer another lower toxicity alternative; however, these formulations face challenges in encapsulating negatively charged TNAs due to electrostatic repulsion. To improve encapsulation, cations such as calcium can be incorporated during liposome formation, resulting in the generation of calcium phosphate nanoparticles that reduce nucleic acid repulsion [79]. Calcium-phosphate anionic liposomes loaded with siRNA have demonstrated more effective endosomal escape than the cationic transfection reagent lipofectamine 6000 [80]. Perhaps more importantly, in this study, toxicity was significantly reduced with this anionic delivery formulation compared to lipofectamine during longer transfection durations.

Lipid-based delivery vehicles remain a promising approach for enhancing the delivery of TNAs by improving stability, uptake, and endosomal escape. In addition to cationic, neutral, anionic, or ionizable lipids, helper lipids such as cholesterol, phospholipids, and PEGylated lipids play an important role in improving escape efficiency [81]. These helpers influence the positioning of nanoparticle lipids, promoting more effective interactions with cellular membranes and facilitating endosomal escape [82]. Ongoing research into optimizing LNP composition holds potential to further enhance delivery efficacy while minimizing associated toxicity.

#### 3.3.2. Exogenous Stimuli Responsive Delivery Vehicles

In addition to endogenous factors like pH, endosomal escape can be triggered by external stimuli such as light, magnetism, and ultrasound, offering a more controlled approach to enhance the delivery of therapeutics. One of the most well-studied methods, photochemical internalization (PCI), uses photosensitive molecules that produce reactive oxygen species (ROS) upon exposure to light [83]. Following endocytosis, light stimuli induce ROS formation causing the perforation of the endosomal membrane and subsequent therapeutic release. Specifically, the formation of a highly reactive singlet oxygen which readily oxidizes biological materials has been shown to be essential for this PCI-mediated endosomal escape [84].

A second light-activated method involves a photothermal mechanism, which utilizes photothermally responsive materials, such as gold nanoparticles (AuNPs), to achieve endosomal escape. When irradiated, AuNPs absorb light and generate localized heat, causing the formation of vapor nanobubbles in the surrounding fluid [85]. These nanobubbles physically disrupt the endosomal membrane, facilitating the release of therapeutic cargo. Other photothermal materials such as melanin-poly-l-lysine polymers have also been employed to enhance the delivery of siRNA via this mechanism [86].

Magnetic fields offer another strategy for inducing endosomal escape. Magnetic nanoparticles, typically composed of iron oxide, respond to an alternating magnetic field by generating heat or physically rotating [87,88]. This heating effect, similar to the photothermal mechanism, destabilizes the membrane and promotes the release of cargo. Alternatively, the mechanical disruption caused by nanoparticle rotation can also facilitate escape. This approach has been successfully applied in the delivery of siRNA using iron oxide nanocages [89].

Stimulation with ultrasound also provides a promising technique for enhancing endosomal escape, primarily through cavitation, a process in which ultrasound-responsive materials such as gas-filled microbubbles expand and collapse in response to focused ultrasound waves [90]. The mechanical forces generated by this cavitation can disrupt the endosomal membrane, allowing for the release of therapeutic cargo. Co-delivery of DNA complexes with microbubbles has demonstrated increased transfection efficacy following ultrasound exposure [91]. Other studies suggest that ultrasound alone, without the need for microbubbles, may be sufficient to permeabilize membranes and improve delivery efficiency, though the exact mechanisms remain under investigation [92].

The use of exogenous stimuli, including light, magnetic fields, and ultrasound, provides a method for controlled, localized, and on-demand induction of endosomal escape for TNA delivery. Magnetic and ultrasonic approaches offer the advantage of deeper tissue penetration compared to light-based methods, which are limited to a few millimeters in biological tissues [93]. Despite their potential, these strategies rely on the physical perturbation of the endosomal membrane, which can pose challenges related to tissue damage or off-target effects. Optimizing stimulation parameters is a key focus for making these non-invasive methods safer and more efficient for therapeutic delivery.

## 4. Alternative Strategies to the Endosomal Escape Problem

Despite numerous advancements in endosomal escape research, no single approach has yet proven to be both highly effective and safe for widespread clinical application. This underscores the need for novel, innovative solutions to overcome the challenges of endosomal entrapment. In this section, we explore two promising alternative strategies: bypassing the endosomal pathway via direct cytosolic delivery, and the targeted manipulation of intracellular trafficking pathways to reroute TNAs for more efficient delivery (Figure 3). These approaches hold potential to sidestep the limitations of current endosomal escape methods while improving therapeutic efficacy and safety.

### 4.1. Direct Cytosolic Entry

A logical approach to preventing endosomal entrapment is to bypass the endo-lysosomal system entirely by delivering TNAs directly into the cytosol. While endocytosis is the dominant pathway of cellular uptake, direct translocation across the plasma membrane can also occur. One method to differentiate between endocytosis and membrane fusion is by conducting transfection experiments at varying temperatures. Since endocytosis is energy-dependent, lowering the temperature should inhibit this process. However, some LNPs have been shown to enter cells even at 4 °C, suggesting that these nanoparticles can fuse directly with the plasma membrane, releasing their cargo into the cytosol through an energy-independent mechanism [94]. Similar results have been observed with siRNA delivered via lipoplexes, where approximately 5% of uptake occurs through endocytosis-independent pathways [95]. Moreover, this study found inhibiting endocytosis does not significantly reduce siRNA activity, indicating that membrane translocation could be the primary route of productive delivery. Building on this insight, recent delivery formulations have been designed specifically to leverage plasma membrane fusion for enhanced cytosolic entry.

One approach to enhancing direct cytosolic delivery involves modifying LNPs with coiled-coil peptides that mimic the function of SNARE proteins. SNAREs are endogenous proteins responsible for promoting vesicle fusion with plasma membranes in various biological processes [96]. Upon the binding of complementary SNARE proteins, spontaneous energy-independent membrane fusion occurs. This mechanism was mimicked to deliver mRNA using LNPs modified with the coiled-coil lipopeptide CPE4 [97]. The CPE4 platform demonstrated enhanced gene delivery even in the presence of endocytosis inhibitors, suggesting direct fusion with the plasma membrane.

In addition to fusogenic proteins, peptides rich in arginine residues have been found to directly penetrate the plasma membrane. This capability is attributed to the presence of guanidinium groups, with guanidinium-modified CPPs having delivered ASOs into cells via direct translocation [98]. The role of arginine in promoting cytosolic delivery has also been extended to various nanoparticle formulations. For example, quantum dot nanoparticles terminated with arginine have been demonstrated to cross the plasma membrane via direct translocation [99]. Additionally, arginine-containing polymers can form membrane pores to deliver plasmid DNA (pDNA) directly into the cytosol [100]. While some designs effectively leverage this mechanism for delivery, not all arginine-containing nanoparticles easily enter cells. Key factors such as nanoparticle size and hardness play critical roles in determining the specific uptake pathway [101,102].

### 4.2. Manipulation of Intracellular Trafficking

To explore potential enhancers of TNA delivery, numerous small molecule screenings have been conducted to assess their impact on therapeutic efficacy. While some small molecules improve TNA effectiveness by disrupting the endosomal membrane as previously discussed, other studies have identified small molecules that enhance their activity through different mechanisms, particularly by modifying intracellular trafficking pathways [103]. This suggests that targeting intracellular trafficking may offer an alternative approach to overcoming the challenges of endosomal escape.

Following endocytic uptake, one potential trafficking pathway involves the recycling of endosomes back to the plasma membrane, where their contents are expelled via exocytosis. One study found that up to 70% of siRNA delivered by LNPs is expelled through this recycling process, suggesting that inhibiting endosomal recycling may enhance delivery efficiency [17]. Initial evidence for this approach focused on the protein Niemann-Pick type C1 (NPC-1), a key regulator of the endosomal recycling pathway. NPC-1 knockout cells have been shown to retain LNPs in endosomes for a longer duration [17]. Moreover, inhibiting NPC-1 with the small molecule NP3.47 has been shown to significantly increase siRNA accumulation in endosomes, resulting in enhanced gene silencing [104]. Although this strategy does not directly improve endosomal escape, retaining cargo inside the cell longer could increase the probability of escape events.

Recently, the cytosolic delivery of ASOs was shown to increase from 3% to 10% when co-administered with the synthetic sphingolipid analog SH-BC-893 [43]. SH-BC-893 inhibits the small GTPase ARF-6, which in its active form promotes the movement of endosomes to the plasma membrane, further supporting the idea that limiting endosomal recycling can improve delivery [105]. Interestingly, SH-BC-893′s effect is not limited to recycling inhibition; it also prevents lysosomal fusion. This dual action, blocking both recycling and degradation pathways, provides a greater opportunity for ASOs to escape endosomes. Other studies have reported similar findings, with molecules such as ammonium ions used to inhibit lysosomal fusion, resulting in significantly increased ASO activity [106].

Small molecules like Retro-1, an inhibitor of endo-Golgi retrograde transport, have also been shown to significantly enhance the endosomal escape and activity of ASOs [107]. Retro-1 was originally identified in a screen for compounds affecting the trafficking of bacterial toxins, many of which use the retrograde transport pathway to move from endosomes to the Golgi and eventually into the cytosol [108]. Blocking this pathway was effective at inhibiting toxin effects, but how this inhibition enhances ASO escape remains unclear. One hypothesis is that preventing retrograde transport retains ASOs in late endosomes, which may be more amenable to cytosolic escape. Another theory suggests that Retro-1 may directly act on late endosomes to promote ASO release, although no evidence of endosomal rupture or pH change has been observed, distinguishing it from small molecule endolytic agents [107]. Interestingly, ASO-Retro-1 conjugates have shown much lower efficacy compared to ASOs administered with Retro-1 separately, suggesting that Retro-1 needs cytosolic access to reach its target effector [109]. Adding to the complexity, different Retro-1 analogs have been found to selectively affect either ASOs or toxins, with only Retro-1 impacting both [110]. This implies that, while similar, ASO and toxin trafficking mechanisms differ in key aspects, highlighting the specificity of their action in ASO trafficking.

While these findings emphasize the significant role of intracellular trafficking in TNA delivery efficiency, targeting key peptide mediators in these pathways poses challenges. Many such proteins are not easily druggable, limiting the ability to specifically manipulate them [111]. Furthermore, vesicle and protein interactions serve essential physiological functions that could be disrupted by chemical interventions. For instance, NPC-1 inhibition, although beneficial for prolonging TNA retention in endosomes, is not ideal therapeutically due to its association with lysosomal lipid storage disorders [112]. Similarly, impaired lysosomal degradation is linked to numerous diseases, as the inability to clear cellular waste products can lead to harmful accumulations [113]. It remains unclear whether the temporary inhibition of these pathways via pharmacological agents would cause long-term adverse effects, but this remains an important consideration. Ultimately, examples of direct cytosolic delivery and intracellular trafficking manipulation have demonstrated promising results for enhancing TNA delivery. While these alternative strategies may not entirely solve the endosomal escape problem, when used in conjunction with traditional endosomal escape platforms, they could offer greater therapeutic potential.

## 5. Nuclear Localization of TNAs

While endosomal escape is often regarded as the major bottleneck in TNA delivery, it is not the final hurdle. For SSOs, nuclear localization is crucial to modulate splicing as pre-mRNA is located in the nucleus. Furthermore, RNaseH and RNAi mechanisms can induce the degradation of both cytoplasmic and nuclear targets [23,24]. In fact, subcellular fractionation experiments show a direct correlation between the nuclear concentration of TNAs and the extent of mRNA knockdown; however, most remain in the cytosol after transfection [114]. The cytoplasmic retention of TNAs exposes them to degradation by intracellular nucleases before interaction with target mRNA, indicating the importance in rapid nuclear accumulation following endosomal escape [10].

Numerous factors influence the nuclear localization of TNAs, particularly through protein interactions. One of the earliest structural modifications made to ASOs, the PS backbone, significantly enhances their activity, potentially due to an increased binding affinity for cellular proteins that regulate various ASO functions [115]. A key aspect of nuclear localization is the formation of ASO aggregates known as PS-bodies. These small, membrane-less structures form in the nucleus after ASO delivery and are induced by interactions between ASOs and proteins, which act as nucleation sites for aggregation [116]. Although the exact role of PS-bodies remains unclear, they are believed to serve as storage sites for ASOs, RNA, and associated proteins. Some studies suggest that PS-bodies do not significantly impact ASO activity, while others indicate that they may play a crucial role in nuclear localization. For instance, Liang et al. demonstrated that the β subunit of chaperonin T-complex 1 (TCP-1β) associates with ASOs inside PS-bodies and enhances their activity [117]. Additionally, the nuclear localization of ASOs and PS-bodies appears to depend on the function of Ran GTPase, indicating an active transport mechanism. Ultimately, while the significance of PS-body formation in the nuclear localization and activity of ASOs is currently an open question, it remains an interesting area for further research.

Intracellular trafficking pathways can also play a significant role in nuclear localization. Research indicates that ASO activity is more pronounced in cell types with efficient trafficking to the nuclear region [118]. This likely occurs because the localization of ASO-containing endosomes to perinuclear regions allows escaped ASOs to enter the nucleus more rapidly, thereby enhancing their functional capacity. The migration of endosomes toward the nucleus, facilitated by transport along microtubules, has been found to depend on various endoplasmic reticulum (ER) tethering proteins, including RNF26, SQSTM1, and UBE2J1 [119]. These interactions suggest that the connectivity between endosomes and other organelles may mediate effective nuclear transport. Conversely, the accumulation of lysosomes in perinuclear regions has been linked to increased ASO degradation, which ultimately reduces their activity [120]. This highlights the importance of not only the trafficking pathways themselves but also the spatial organization of organelles in regulating the nuclear localization of TNAs.


*Nuclear Localization Signals*


Compared to endosomal escape, far fewer strategies have been explored to enhance the nuclear localization of TNAs. The primary method investigated involves the use of nuclear localization signals (NLSs), which are short, predominantly cationic peptide sequences that facilitate the transport of proteins into the nucleus [121]. NLS sequences are found in various viral and endogenous proteins that play crucial roles in physiological processes such as transcription regulation and cell cycle control. This molecular mechanism of nuclear transport heavily relies on a family of transport proteins known as importins. Upon binding to an NLS, importins shuttle proteins into the nucleus through the nuclear pore complex via an active transport process mediated by Ran GTPase [122] (Figure 4).

The application of NLSs in drug delivery has primarily focused on larger molecules like proteins or pDNA, due to their inability to passively cross the nuclear membrane, compared with smaller TNAs which can more readily traverse this barrier [123]. However, the penetration of TNAs into the nucleus can still be restricted by factors such as their size, secondary structure, and the potential need for active transport [32]. Moreover, the permeability of the nuclear pore complex can vary based on cell type and environmental conditions, further influencing entry [124].

The ability of NLSs to promote nuclear transport has made them a promising focus for enhancing TNA delivery and overcoming barriers to effective gene therapy. Similarly to the previously discussed CPP conjugates, TNAs can also be conjugated to NLS peptides to improve their nuclear uptake. Early studies in this area explored the conjugation of ASOs to NLSs derived from various sources, including the SV40 T-antigen, influenza virus nucleoprotein, and HIV-1 TAT protein [22]. A key finding from Kubo et al. demonstrated that unmodified ASOs were evenly distributed throughout the cytoplasm and nucleus, with similar localization patterns observed for ASOs conjugated to a control peptide. In contrast, NLS-ASO conjugates exhibited a marked preference for nuclear localization. This enhanced nuclear accumulation significantly increased antisense activity, with the most effective NLS from the SV40 T-antigen boosting treatment efficacy from 87.6% to 99.6% within 24 h. The SV40 T-antigen was the first NLS identified, with its corresponding peptide sequence PKKKRKV now recognized as a prototypical NLS [121]. Beyond enhancing nuclear localization, the cationic nature of NLS peptides may also improve cellular uptake through mechanisms similar to those of cationic CPPs, potentially facilitating better overall delivery.

Additionally, combining NLSs with other delivery strategies, such as nanoparticle encapsulation or lipid-based systems, has the potential to yield synergistic effects that maximize the therapeutic efficacy of TNAs. For example, Tarvirdipour et al. designed a peptide-based nanoparticle functionalized with the NLS sequence KRKR specifically to deliver ASOs targeting the Bcl-2 gene [125]. This innovative delivery system demonstrated a notable increase in the association of nanoparticles with the nuclear membrane, as well as a higher proportion of nanoparticles entering the nucleus compared to those lacking the NLS. Importantly, this enhanced nuclear accumulation corresponded with a significantly greater knockdown of the target gene. In a separate study, the incorporation of an NLS into poly-lactic-co-glycolic acid (PLGA) polymers not only enhanced nuclear localization but also improved nanoparticle loading, suggesting that NLSs can confer additional stability benefits to the nanoparticles themselves [126].

Despite the potential benefits of incorporating NLSs into TNA delivery systems, a recent study has revealed some contrasting findings regarding their effectiveness. Among seven different NLSs tested, only one sequence, SKKKKTKVC derived from the transcription factor TFIIE-β, successfully enhanced the nuclear localization of ASOs [127]. However, this increased localization did not translate into enhanced antisense activity. The researchers attributed this discrepancy to endosomal entrapment in the perinuclear region, noting that treatment with the endolytic small molecule chloroquine restored proper splice modulation. These findings underscore that while nuclear localization is crucial, endosomal escape remains a significant barrier to effective TNA delivery. Combinatorial approaches, such as pairing an NLS with fusogenic peptides, may provide a dual mechanism to enhance treatment efficacy through the promotion of both nuclear entry and endosomal escape [128]. By targeting both pathways, such strategies could lead to a more comprehensive enhancement of TNA activity.

## 6. Conclusions and Future Perspectives

TNAs including ASOs and siRNA hold immense potential to transform the treatment landscape for various conditions by directly targeting disease-causing genes. However, their therapeutic efficacy is significantly hindered by challenges such as endosomal entrapment and inefficient nuclear localization. Despite the numerous strategies proposed to mitigate these issues, no single platform has emerged as a comprehensive solution or achieved clinical applicability. To address these limitations, we advocate for focused research in several key areas:

Cell type and cargo-specific understanding of intracellular trafficking: A critical aspect of enhancing endosomal escape lies in thoroughly understanding how TNAs are trafficked within cells and their natural mechanisms for escaping endosomal entrapment. Although recent studies have improved our comprehension of these processes, emerging evidence indicates that trafficking mechanisms can vary significantly based on cell type and the specific cargo involved. For example, while the small molecule retro-1 enhances ASO efficacy, it has no effect on siRNA delivery [107]. Additionally, different endosomal escape enhancers have demonstrated varying effectiveness when ASOs are delivered via lipid nanoparticles compared to cholesterol conjugated forms [103]. Therefore, further research tailored to the endosomal escape mechanisms of specific cargos (ASOs vs. siRNA), delivery systems (free nucleotides, nanoparticle-encapsulated, and peptide-conjugated formats), and the relevant cell types is essential for the practical application of these approaches.

Focus on non-toxic endosomal escape strategies: One of the most significant challenges regarding endosomal escape is achieving a balance between escape efficiency and cytotoxicity. The endo-lysosomal system plays a crucial role in intracellular trafficking and the clearance of cellular waste products. Disrupting these pathways can lead to cellular stress and detrimental effects. For instance, the formation of breaks in the endosomal membrane can trigger a membrane damage response mediated by galectins, initiating downstream inflammatory pathways that can cause cytotoxicity [129]. Consequently, strategies that avoid physically damaging the endosomal membrane such as promoting membrane fusion or facilitating vesicle budding/collapse, are the most promising for minimizing potential toxicity, warranting further investigation.

Alternative and combinatorial strategies: Current delivery approaches alone are insufficient to overcome the rate-limiting barriers associated with TNA delivery. We propose expanding research efforts to explore combinatorial strategies that integrate multiple techniques to optimize these processes. For instance, the synergistic use of nanoparticles and peptides in conjunction with small molecule modulators of intracellular trafficking could significantly enhance delivery efficiency. While improving endosomal escape is crucial, the significance of efficient nuclear localization must also be emphasized. Nuclear accumulation is essential for the function of certain TNAs and exploring innovative strategies to facilitate this process could dramatically improve therapeutic outcomes. Incorporating NLS peptides into existing delivery platforms presents a promising avenue to tackle these challenges.

In summary, addressing the barriers to effective TNA delivery necessitates a multifaceted approach that prioritizes understanding intracellular trafficking mechanisms, developing non-toxic escape strategies, and exploring innovative combinatorial methods. Continued research in these areas could pave the way for more effective nucleic acid-based therapies, especially for diseases where precise gene expression regulation is critical. By overcoming these challenges, TNAs can realize their full therapeutic potential, ultimately benefiting a wide range of patients.

## Figures and Tables

**Figure 1 molecules-29-05997-f001:**
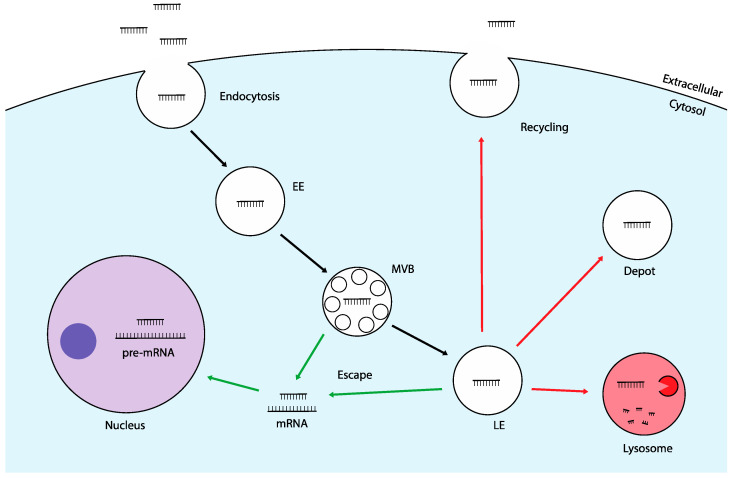
Uptake and intracellular trafficking of TNAs. Following endocytosis, TNAs become encapsulated inside early endosomes (EE) which undergo maturation to multivesicular bodies (MVBs) and late endosomes (LEs). Non-productive pathways (red) do not permit TNAs to reach their intracellular targets. Such pathways include recycling to the plasma membrane, retention in depot endosomes, or enzymatic degradation in lysosomes. Productive pathways (green) allow the successful escape of TNAs into the cytosol to interact with mRNA targets, or eventually the nucleus when targeting pre-mRNA. A small portion (1–2%) of freely delivery TNAs escape endosomes during trafficking primarily from MVBs and LEs.

**Figure 2 molecules-29-05997-f002:**
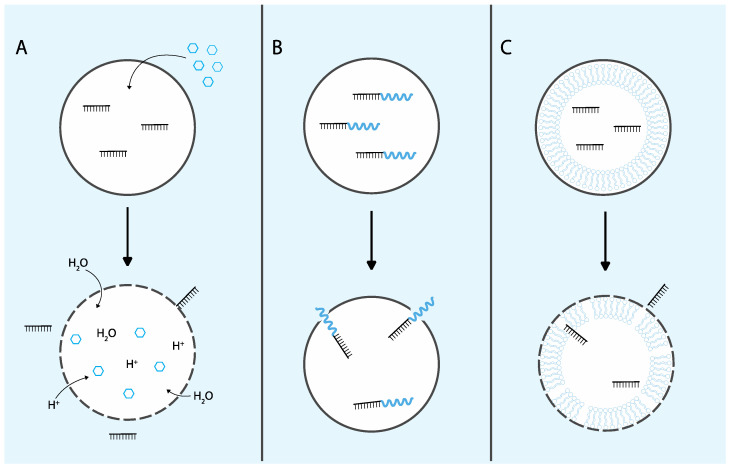
Traditional endosomal escape strategies. (**A**) Cationic amphiphilic small molecules (CADs) such as chloroquine enter endosomes buffering changes in pH. The resulting proton-sponge effect induces osmotic swelling and endosomal rupture, allowing TNAs to escape. Other small molecules may directly interact with endosomes causing membrane destabilization. (**B**) Peptide-mediated endosomal escape can be facilitated by biomimetic or cell-penetrating peptides. Interaction between cationic peptides and the anionic endosomal membrane causes fusion, membrane destabilization, or pore formation. (**C**) Non-viral delivery vehicle-mediated endosomal escape can be achieved using lipid nanoparticles. Cationic or ionizable lipids facilitate fusion with the endosomal membrane, allowing TNA release.

**Figure 3 molecules-29-05997-f003:**
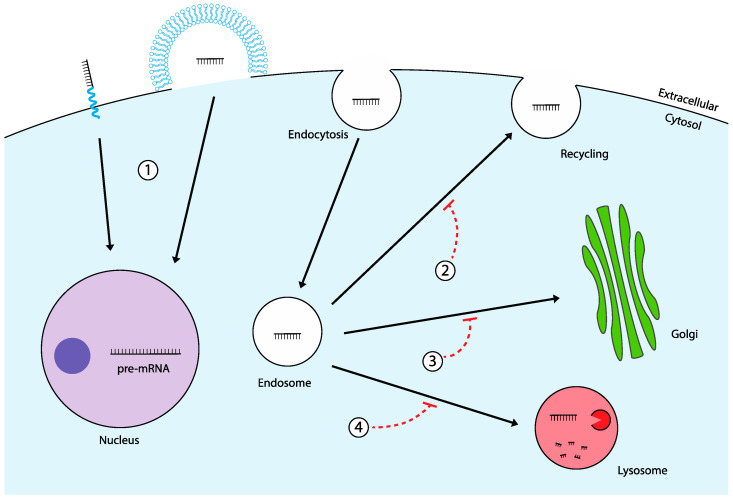
Alternative strategies to the endosomal escape problem. (1) Direct cytosolic entry of TNAs which can be facilitated by peptide or nanoparticle-mediated delivery. Avoiding endocytosis and subsequent endosomal entrapment allows free translocation to the nucleus. (2) Inhibition of endosomal recycling can be accomplished through small molecules such as NP3.47. Preventing the exocytosis of internalized TNAs provides increased potential for endosomal escape events. (3) Inhibition of endo-Golgi retrograde transport with small molecules such as Retro-1. The mechanism of action remains unclear but may increase the retention of TNAs in endosomes, increasing the probability for escape. (4) Inhibition of endo-lysosomal fusion with molecules such as SH-BC-893. Preventing the degradation of TNAs entrapped in endosomes increases their cytosolic quantity, improving treatment efficacy.

**Figure 4 molecules-29-05997-f004:**
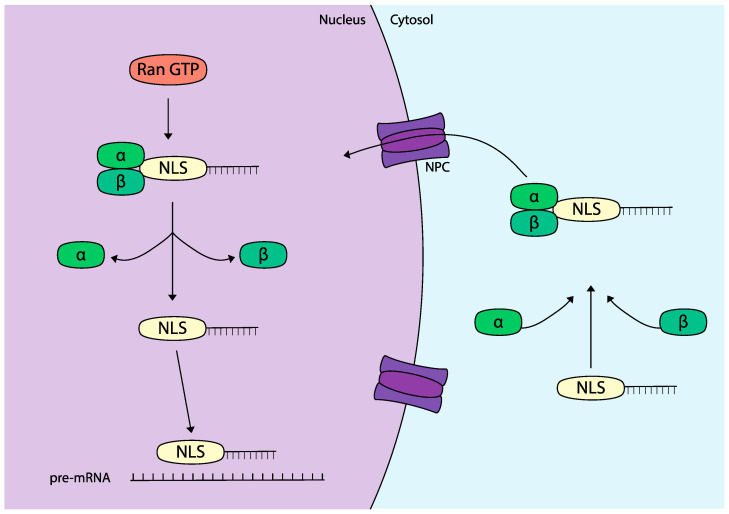
Mechanism nuclear localization signal (NLS) internalization. TNA-NLS peptide conjugates interact with importin-α and β in the cytosol which facilitate active transport through the nuclear pore complex (NPC). Following nuclear entry, the binding of RanGTP causes the dissociation of the complex. The free TNA-NLS is now capable of binding to target pre-mRNA in the nucleus.

## Data Availability

No new data were created or analyzed in this study. Data sharing is not applicable to this article.

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
