# Peer review of "Endosomal Escape and Nuclear Localization: Critical Barriers for Therapeutic Nucleic Acids"

_molecules, 2024, doi:10.3390/molecules29245997_

Round 1

Reviewer 1 Report

Comments and Suggestions for Authors

This review explores the current understanding of endosomal escape, and nuclear transport of ASOs. Furthermore, strategies aimed at overcoming these challenges, including the use of endosomal escape agents, peptide-ASO conjugates, non-viral delivery vehicles, and nuclear localization signals, have been examined. By improving both endosomal escape and nuclear localization, significant advances in ASO-based therapeutics can be realized, ultimately enhancing their potency and expanding their clinical utility. It's suggested that papers related to ASOs and siRNA encapsulated with DNCA/CLD etc should be reviewed either in this paper.  

Author Response

Reviewer 1:

This review explores the current understanding of endosomal escape, and nuclear transport of ASOs. Furthermore, strategies aimed at overcoming these challenges, including the use of endosomal escape agents, peptide-ASO conjugates, non-viral delivery vehicles, and nuclear localization signals, have been examined. By improving both endosomal escape and nuclear localization, significant advances in ASO-based therapeutics can be realized, ultimately enhancing their potency and expanding their clinical utility. It's suggested that papers related to ASOs and siRNA encapsulated with DNCA/CLD etc should be reviewed either in this paper.  

  • Thank you for the feedback. We agree that DNCA/CLD represent a promising delivery strategy for ASOs/siRNA. This was mentioned in section 3.3.1 lipid delivery vehicles (line 390)

Reviewer 2 Report

Comments and Suggestions for Authors

The authors discuss two issues important for therapeutic nucleic acids (TNAs) applications, TNA escaping from endosomes and nuclear localization of TNAs. This is a valuable review however, it would be to the benefit of the reader if the issues raised were discussed in a broader perspective.

In addition to ASO which is the nominal subject of the review (based on the title), there are other TNAs such as siRNA and others. They all are short fragments of DNA or RNA, the primary mechanism of action for most TNAs at some stage is similar - through the complementarity of their nucleobase sequence to the nucleic acid target, also their mode of administration is analogous. They also share common problems such as stability in biological environment and delivery issue including endosomal escape and sometime nuclear localization.

However, reading the text of the manuscript one can get the impression that the authors consider all types of therapeutic nucleic acids as “antisense therapeutics”, ASO, even writing about RNA therapeutics (line 199, 265, 392, 397 and others), which is confusing.

This problem could be avoided by using the broader term “therapeutic nucleic acids” (TNA) instead of ASO. And use the term and abbreviation ASO only when the authors are actually writing about antisense oligodeoxynucleotides.

The statement “two primary mechanisms of action (of ASO): RNase H-mediated degradation or splice switching” (line 34) is inaccurate, in reality the two main mechanisms of ASO action are RNase H-mediated mRNA degradation or mRNA blockage. Modulation of splicing is only a consequence of the RNA blockage, and this mechanism can cause other effects than splicing modulation too.

It would also be advisable that when discussing specific literature examples of TNA in the context of endosomal escape or nuclear transport, the authors state what method of TNA delivery was actually used, chemical, physical, nanocarriers, unassisted?

Author Response

Reviewer 2:

The authors discuss two issues important for therapeutic nucleic acids (TNAs) applications, TNA escaping from endosomes and nuclear localization of TNAs. This is a valuable review however, it would be to the benefit of the reader if the issues raised were discussed in a broader perspective.

In addition to ASO which is the nominal subject of the review (based on the title), there are other TNAs such as siRNA and others. They all are short fragments of DNA or RNA, the primary mechanism of action for most TNAs at some stage is similar - through the complementarity of their nucleobase sequence to the nucleic acid target, also their mode of administration is analogous. They also share common problems such as stability in biological environment and delivery issue including endosomal escape and sometime nuclear localization.

However, reading the text of the manuscript one can get the impression that the authors consider all types of therapeutic nucleic acids as “antisense therapeutics”, ASO, even writing about RNA therapeutics (line 199, 265, 392, 397 and others), which is confusing.

This problem could be avoided by using the broader term “therapeutic nucleic acids” (TNA) instead of ASO. And use the term and abbreviation ASO only when the authors are actually writing about antisense oligodeoxynucleotides.

  • Thank you for the constructive feedback. As suggested, the focus of the paper was edited to more broadly address endosomal escape and nuclear localization of therapeutic nucleic acids (TNAs) (entire paper).
  • Antisense oligonucleotides (ASOs) and small interfering RNA (siRNA) are now referenced as two specific examples of TNAs (line 31). Information was also added regarding their respective mechanisms of action (lines 38-45)

The statement “two primary mechanisms of action (of ASO): RNase H-mediated degradation or splice switching” (line 34) is inaccurate, in reality the two main mechanisms of ASO action are RNase H-mediated mRNA degradation or mRNA blockage. Modulation of splicing is only a consequence of the RNA blockage, and this mechanism can cause other effects than splicing modulation too.

  • The two major mechanisms of action for ASOs were edited to more accurately include RNase H-mediated degradation and steric hindrance (line 36) with splice modulation addressed as a specific example of the steric hindrance mechanism.

It would also be advisable that when discussing specific literature examples of TNA in the context of endosomal escape or nuclear transport, the authors state what method of TNA delivery was actually used, chemical, physical, nanocarriers, unassisted?

  • When discussing specific literature on endosomal escape and nuclear localization information regarding the method of delivery was added where applicable (line 143, 204, 225, 255, 268, 335, 541).

Reviewer 3 Report

Comments and Suggestions for Authors

General Comment

The manuscript addresses a topic of great interest today, especially since antisense oligonucleotides have shown their potential clinical utility and therapeutic versatility. However, the efficacy of their use in the clinic represents only a small proportion of their true potential, due to their low bioavailability after administration and the authors make a critical review of the two main causes of this. They focus their attention firstly on the significant loss of functional availability of oligonucleotides during their cellular internalization route with access to the cytoplasm and secondly on the great difficulty in accessing the nucleus where oligonucleotides can best manifest their maximum efficacy, but to cross the nuclear pore the help of appropriate transporters is required. In both cases, the strategies that have been addressed in order to improve the bioavailability of oligonucleotides and thus overcome these limiting barriers to their optimal efficacy are indicated.

Strengths

The objective of the review is very well focused. In each of the proposed routes, the pros and cons of the cellular bioavailability of oligonucleotides at each stage are analyzed, accompanied by the references that support the observations and comments of the reviewers.

The manuscript is easy to read and offers updated information to the average interested reader.

The information is of educational interest to a large number of people interested in new therapeutic tools and in particular nucleic acids and also prepares the reader to better understand the management of these new therapeutic possibilities.

Weaknesses

No comments worth highlighting

Author Response

Reviewer 3:

General Comment

The manuscript addresses a topic of great interest today, especially since antisense oligonucleotides have shown their potential clinical utility and therapeutic versatility. However, the efficacy of their use in the clinic represents only a small proportion of their true potential, due to their low bioavailability after administration and the authors make a critical review of the two main causes of this. They focus their attention firstly on the significant loss of functional availability of oligonucleotides during their cellular internalization route with access to the cytoplasm and secondly on the great difficulty in accessing the nucleus where oligonucleotides can best manifest their maximum efficacy, but to cross the nuclear pore the help of appropriate transporters is required. In both cases, the strategies that have been addressed in order to improve the bioavailability of oligonucleotides and thus overcome these limiting barriers to their optimal efficacy are indicated.

Strengths

The objective of the review is very well focused. In each of the proposed routes, the pros and cons of the cellular bioavailability of oligonucleotides at each stage are analyzed, accompanied by the references that support the observations and comments of the reviewers.

The manuscript is easy to read and offers updated information to the average interested reader.

The information is of educational interest to a large number of people interested in new therapeutic tools and in particular nucleic acids and also prepares the reader to better understand the management of these new therapeutic possibilities.

Weaknesses

No comments worth highlighting

  • Thank you for the positive feedback. As no weaknesses were highlighted there were no revisions made based on reviewer 3’s comments.